# Isolated Epiglottic Manifestations of HIV Infection: Two Cases Reports

**DOI:** 10.3390/microorganisms10122404

**Published:** 2022-12-05

**Authors:** Yi-Chieh Lee, Hsueh-Yu Li, Wan-Ni Lin

**Affiliations:** 1Department of Otolaryngology Head & Neck Surgery, Chang Gung Memorial Hospital at Linkou, 5 Fushing St., Taoyuan 333, Taiwan; 2London School of Hygiene and Tropical Medicine, London WC1E 7HT, UK

**Keywords:** HIV, larynx, epiglottis, epiglottitis, epiglottic tumor

## Abstract

Diagnosis of the Human Immunodeficiency Virus (HIV) remains challenging due to non-specific clinical presentations and mostly flu-like symptoms, e.g., fever, headache, sore throat, and general weakness. Oral lesions, such as oral candidiasis and Kaposi sarcoma, are also frequently associated with HIV infection, whereas laryngeal manifestations are rare. We report two cases of newly diagnosed HIV patients with clinical presentations of sore throat, and endoscopy revealed an epiglottic ulcerative tumor-like lesion. A laryngomicrosurgical biopsy of the lesions was performed for persistent symptoms and suspicion of malignancy. The result revealed acute and chronic inflammation without a conclusive pathology diagnosis. Further laboratory analysis was arranged in consideration of autoimmune diseases, Epstein–Barr virus (EBV), and HIV infection due to their persistent and atypical symptoms. The results were positive for HIV infection. These patients were treated successfully with antiviral treatment and the laryngeal symptoms improved within weeks. In patients with idiopathic and persistent epiglottitis or an epiglottic ulcer after medical treatment, HIV infection needs to be considered as a potential etiology in order to institute proper treatment.

## 1. Introduction

Aphthous ulcer in oral mucosa is a common symptom in the general population and is also one of the most common symptoms in HIV patients [1]. It can be recurrent and persistent in immunocompromised patients and is more likely to be seen in the later stage of HIV infection. Other oral manifestations are also frequently associated with HIV infection, including oral candidiasis, Kaposi sarcoma, cancrum oris, and herpes infection [2]. However, isolated ulcerative lesions over epiglottis are rare in HIV patients and there is a paucity of English literature reviewing laryngeal manifestations in HIV patients [3]. We observed two cases of newly diagnosed HIV patients with clinical presentations of epiglottic lesions.

## 2. Case Presentations

### 2.1. Case Report 1

A 56-year-old male presented to our emergency department due to severe odynophagia for one day. He reported smoking cigarettes occasionally. An otolaryngological examination showed edematous and erythematous changes with ulceration over epiglottis with patent airway.

An initial impression of acute epiglottis was made and he was admitted to the hospital on 23 June 2014, for medication treatment. He was then discharged five days later after the clinical condition became stable. However, the patient returned to the emergency department the next day after discharge with repeated acute onset odynophagia. The fiberscope examination showed diffused granulation tissues over epiglottis, which was suspected of tumor or mycobacterial infection (Figure 1). Laryngomicrosurigical biopsy was arranged along with tuberculosis testing, and the final pathology report showed acute and chronic inflammation, negative for acid-fast stain. With the consideration of immune deficiency or virus infection, blood tests for autoimmune-related antigens and HIV were arranged with a positive HIV Western blot finding and a high level of HIV RNA copy (RNA copy:581,183 copies/mL). He was then transferred to the infection department for further treatment. Esophagogastroduodenoscopy was also arranged without specific findings. Other biochemical tests, including syphilis, hepatitis A virus, hepatitis B virus, and hepatitis C virus, were tested and all came back negative. CD4 T cell number was low (16.0 cells/μL) and, thus, he started antiretroviral treatment (ART) on 8 August 2014, with amivudine 150, zidovudine 300, and Efavirenz 600 mg. His symptoms improved two weeks later after ART.

### 2.2. Case Report 2

A 38-year-old male had a history of tonsillectomy and adenoidectomy due to chronic tonsillitis and hypertrophy of adenoid in 2018. The pathology report showed lymphoid hyperplasia of both tonsils and adenoid. In February 2021, he presented to a local clinic for prolonged sore throat for two months. He had no habit of smoking, drinking, or betelnut chewing. He was referred from the local clinic to our otolaryngology department due to a suspected laryngeal lesion from the local clinic. Fiberscope examination revealed an ulcerative lesion with chondritis over the right epiglottis without the involvement of bilateral vocal cords and their motility (Figure 2). The initial impression was a malignant laryngeal tumor due to its ulcerative appearance. Therefore, laryngomicrosurgery biopsy was arranged and the final histopathological analysis of the biopsy specimen showed neither aberrant expression of lymphoid antigens nor abnormal localization of lymphoid cells, consistent with a reactive process at the morphological and immunohistochemical levels. There were scattered Epstein–Barr virus (EBV)-infected cells seen in the specimen, and a diagnosis of ulcer, granulation tissue, and scattered EBV (+) cells was made. The results of laboratory examination showed microcytic anemia (Hb: 9.6 g/dL, Hct: 31.5%), but otherwise normal findings. The symptom persisted and was refractory to medication. With previous experience dealing with a persistent epiglottic ulcer with inconclusive pathology findings, we arranged further biochemical tests for autoimmune-disease-related antigens, Human Immunodeficiency Virus (HIV) infection, and EBV infection. The results came back as reactive (3385.0) for HIV Ag/Ab combination test (reference range < 1). He was then transferred to the infection department for HIV treatment. Thorough laboratory analysis for the immunological status, hepatitis and syphilis were tested. The patient had low absolute CD4 count (54.0 cells/μL) and latent syphilis (Rapid plasma regain = 1:8; Nontreponemal test > 1:1280). He was treated for latent syphilis and started ART with Lamivudine 300 mg, Dolutegravir 50 mg, and Abacavir 600 mg. A few weeks later, the symptoms gradually resolved. 

## 3. Discussion

HIV infection commonly presents with flu-like symptoms and epiglottis is rarely involved. In this study, we reported two HIV cases with an isolated epiglottic tumor-like lesion, emphasizing the potential etiology of HIV infection in idiopathic and persistent epiglottic lesions. Oral ulcers are commonly found in patients with HIV/AIDS, while epiglottic ulcers mimicking tumor lesions are rarely seen. The management of epiglottic ulcers depends on the patients’ symptoms and characteristics of the lesion. For the persistent non-healing ulcers and the lesion with abnormal characteristics, such as irregular margins, heterogeneous appearances, and destructive structure, a biopsy should be taken to rule out malignancy. For larger masses causing laryngeal obstruction, a biopsy followed by excision should be taken into consideration. Otherwise, close observation with regular fiberscope examination is favored [4]. In addition, acute epiglottitis in HIV patients was not reported until 1989 [5]. A study in New York presented five cases of acute epiglottis in AIDS patients. The disease progression was rapid and devastating in these patients, where some had intubation or even had tracheostomy performed. Due to rapidly progressing airway obstruction, aggressive airway intervention was suggested in the study. Four out of five patients had a previous medical history of a positive HIV antibody test, while one of the patients was diagnosed after treatment for acute epiglottitis. 

Our first patient had recurrent acute epiglottitis in a short period with unusual granulation formation. He had a biopsy over the epiglottic lesion because of its persistence and tumor-like appearance. The pathological examination only showed inflammation. Our second patient presented with a prolonged sore throat, refractory to medication control. Flexible laryngoscopy revealed ulceration with chondritis at the epiglottis. The malignant lesion was first impressed due to its persistency. Inconclusive results from the histopathological analysis made the treatment more difficult.

Both patients had an unusual presentation of the epiglottic lesion with poor response to conservative treatment and, therefore, malignancy and unusual infection were suspected. However, an inconclusive pathology report occurred in both cases, and an immuno-deficiency-related condition was then considered. Their symptoms resolved after starting antiviral treatment. 

According to the recommendations of HIV testing in indicator diseases, in which the prevalence of undiagnosed HIV infection is greater than 0.5%, head and neck lesions include oral hairy leukoplakia, unexplained lymphadenopathy, and unexplained oral candidiasis [6]. Laryngeal manifestations are not listed. However, people infected with HIV are prone to unusual infections and the larynx is also a site of involvement. Reviewing the literature, when involved with the larynx, the accurate diagnosis is crucial for prompt and effective treatment since the larynx is a critical anatomic location and the disease progression might be fast and devastating [5]. Definite diagnosis remains challenging and a biopsy with culture should be obtained to establish the diagnosis. Apart from that, when the laryngeal lesions do not respond to conservative treatment, neoplasm should also be considered and a biopsy should be arranged. To sum up, when a patient presents with persistent symptoms refractory to conservative measures, a biopsy should be conducted to make a diagnosis between unusual infection and neoplasm. When the diagnosis of an unusual infection or inconclusive pathology report is made, HIV infection should be tested.

## 4. Conclusions

From a clinical viewpoint, it is important to consider that immunodeficiency conditions, such as HIV infection, may be the underlying etiology in unusual laryngeal lesions mimicking a laryngeal tumor or epiglottitis. Doctors must be familiar with decision tree and differential diagnosis when patients present with laryngeal complaints refractory to conservative treatments.

## Figures and Tables

**Figure 1 microorganisms-10-02404-f001:**
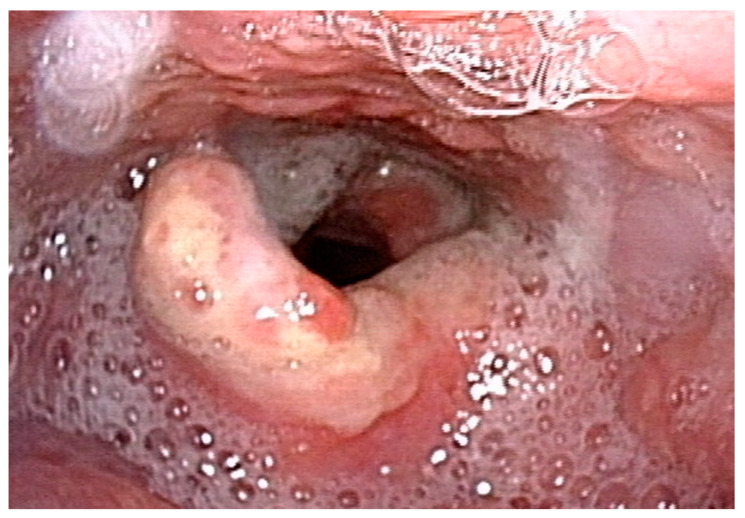
The 56-year-old male presented with granulation tissues over epiglottis.

**Figure 2 microorganisms-10-02404-f002:**
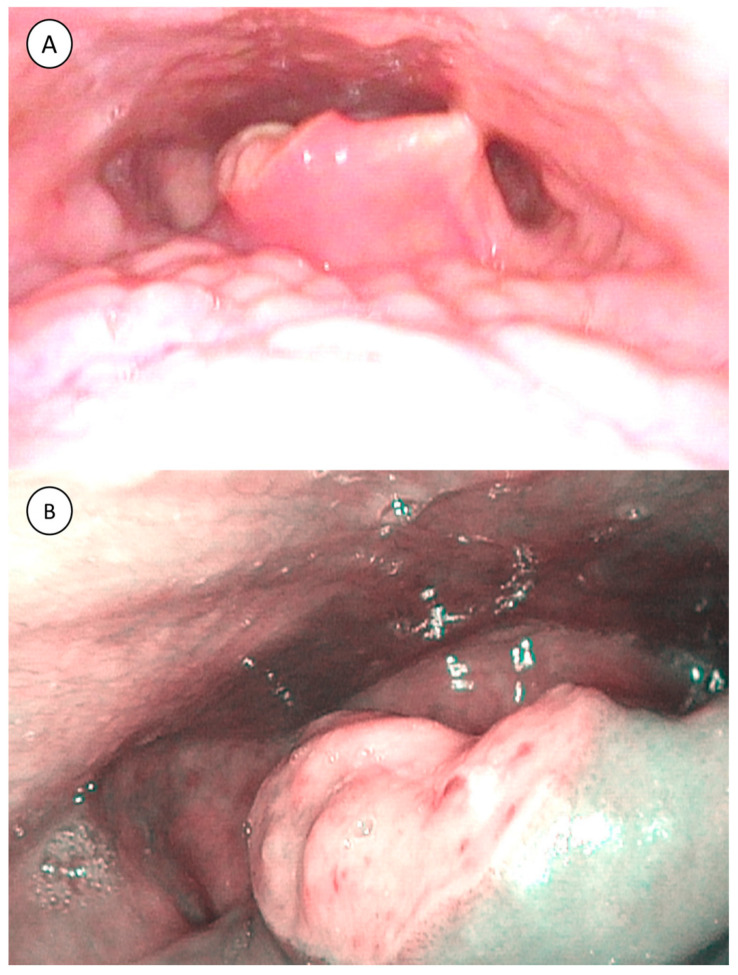
(**A**) The 38-year-old male presented with ulceration with chondritis over the right epiglottis; (**B**) right epiglottic ulcer in narrow-band imaging.

## Data Availability

The data are not publicly available due to the regulation of our institution and the protection of patients’ privacy. However, the data presented in the study are available on request from the corresponding author for further research.

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
