# Peer review of "Isolated Epiglottic Manifestations of HIV Infection: Two Cases Reports"

_microorganisms, 2022, doi:10.3390/microorganisms10122404_

Round 1

Reviewer 1 Report

It is a interesting paper presenting patients with an unusually HIV onset

Did the patients made test for syphilis or other STDs ?

Reviewer 2 Report

Case presentations should be improved. A lot of data concerning eg differential diagnosis in terms of bacterial and viral infections are lacking; it looks that the tests were not done; there is no information about CD4 T cell count in the second pt; There is no information about the time of improvement after ART introduction. There is no information about esophagoscopy, risky behaviours, STIs etc. 

Round 2

Reviewer 2 Report

The authors still did not improve enough case reports descriptions,

Important information is lacking: in both cases which sexually transmitted diseases were excluded should be mentioned; the authors should not use trade names of antiretroviral drugs; according to HIV the drugs name is antiretroviral; 

number of CD4 T cell counts are mentioned without units; the same concerning HIV RNA;

regarding these two cases the same medical procedures indicating the reason of the symptoms should be used; the authors did not convince the reviewer that there can be another reasons of the disease. 

Author Response

Response: Thanks for the comment.

The descriptions about further biochemical tests for the two cases were modified.

The antiretroviral medication was revised. (Page2, 3)

The unit for cell counts and viral loads were added (Page 2)

The case presentations were further revised to explain our clinical decisions and differential diagnosis in managing these two patients.